# Diversity and Distribution of Viruses Infecting Wild and Domesticated *Phaseolus* spp. in the Mesoamerican Center of Domestication

**DOI:** 10.3390/v13061153

**Published:** 2021-06-16

**Authors:** Elizabeth Chiquito-Almanza, Juan Caballero-Pérez, Jorge A. Acosta-Gallegos, Victor Montero-Tavera, Luis Antonio Mariscal-Amaro, José Luis Anaya-López

**Affiliations:** 1Biotechnology Department, National Institute for Forestry Agriculture and Livestock Research (INIFAP), Celaya, Guanajuato 38110, Mexico; ely_sayra@hotmail.com (E.C.-A.); montero.victor@inifap.gob.mx (V.M.-T.); 2Faculty of Chemistry, Autonomous University of Querétaro, Santiago de Querétaro 76017, Mexico; jcaballero@uaq.mx; 3Bean Breeding Program, National Institute for Forestry Agriculture and Livestock Research (INIFAP), Celaya, Guanajuato 38110, Mexico; acosta.jorge@inifap.gob.mx; 4Forestry and Plant Protection Program, National Institute for Forestry Agriculture and Livestock Research (INIFAP), Celaya, Guanajuato 38110, Mexico; mariscal.luis@inifap.gob.mx

**Keywords:** *Phaseolus* spp., potyvirus, carlavirus, begomovirus, alphaendornavirus, cytorhabdovirus, sRSA

## Abstract

Viruses are an important disease source for beans. In order to evaluate the impact of virus disease on *Phaseolus* biodiversity, we determined the identity and distribution of viruses infecting wild and domesticated *Phaseolus* spp. in the Mesoamerican Center of Domestication (MCD) and the western state of Nayarit, Mexico. We used small RNA sequencing and assembly to identify complete or near-complete sequences of forty-seven genomes belonging to nine viral species of five genera, as well as partial sequences of two putative new endornaviruses and five badnavirus- and pararetrovirus-like sequences. The prevalence of viruses in domesticated beans was significantly higher than in wild beans (97% vs. 19%; *p* < 0.001), and all samples from domesticated beans were positive for at least one virus species. In contrast, no viruses were detected in 80–83% of the samples from wild beans. The *Bean common mosaic virus* and *Bean common mosaic necrosis virus* were the most prevalent viruses in wild and domesticated beans. Nevertheless, *Cowpea mild mottle virus*, transmitted by the whitefly *Bemisia tabaci*, has the potential to emerge as an important pathogen because it is both seed-borne and a non-persistently transmitted virus. Our results provide insights into the distribution of viruses in cultivated and wild *Phaseolus* spp. and will be useful for the identification of emerging viruses and the development of strategies for bean viral disease management in a center of diversity.

## 1. Introduction

Virus diseases are a significant threat to agriculture production. The emergence of indigenous and newly introduced viruses is a potential threat to biodiversity in centers of origin and plant domestication, reviewed in [1]; therefore, studies on the dynamics of pathogens and their hosts at sites where wild and domesticated hosts coexist are extremely valuable. Understanding the diversity and evolution of the prevalent viruses in wild and domesticated *Phaseolus* spp. is needed to quantify the impact of viruses on bean cultivation and the effect of human management on population heterogeneity, biodiversity loss and disease risk. Common bean (*Phaseolus vulgaris* L.) is likely the most susceptible plant species in the *Leguminosae* to virus infection [2] and one of the most widely cultivated pulses in the world [3].

There are two major genetic pools of common bean, the Mesoamerican and Andean, reviewed in [4]. Genomic and phenotypic data support the Mesoamerican origin of the common bean (*P. vulgaris* L.) [5,6], with the most likely center of origin in Mexico [6], and one of the Mesoamerican centers of domestication (MCD) is located in the Lerma-Santiago basin in the states of Jalisco and Guanajuato, Mexico [7]. In Mexico, there have not been comprehensive studies on the distribution and identification of viral species that infect domesticated and wild beans. This information is required for strategic breeding and use of cultivars, for informed decision-making on imposing quarantine, and for developing diagnostic tools and predicting the emergence of new viral strains [8]. 

The analysis of RNA sequences from plant samples through next-generation sequencing (NGS) is an effective method for virus detection, reviewed in [9]. Among NGS approaches, small RNA sequencing and assembly (sRSA) offers a feasible alternative for the simultaneous detection of multiple known and novel DNA and RNA viruses without prior knowledge or the need for antibodies or virus-specific primers or probes and is 10-times more sensitive than reverse-transcription real-time PCR (RT-qPCR) [10,11]. The sRSA is based on RNA interference (RNAi) of the plant antiviral defense system, which targets double-stranded RNA (dsRNA). Since DNA viruses also induce the production of small interfering RNA (siRNA) in plants, the small RNA (sRNA) from infected plant tissues can be extracted and sequenced, and the sequences of virus-derived from small interfering RNAs (vsiRNA) can then be reassembled in silico into partial or full viral genomes of both DNA and RNA viruses, which allows their identification by comparison with viral sequences available in databases [10]. 

In aims to expand the knowledge on the diversity, distribution, and dissemination of viruses infecting domesticated and wild beans in the MCD and surrounding areas, symptomatic and asymptomatic samples from field-grown common bean and wild bean populations were collected in Guanajuato, Jalisco, and Nayarit, Mexico, and were analyzed through the sRSA approach using the Illumina platform. The data obtained revealed the presence of nine known viral species, two putative novel alphaendornaviruses, and five virus-like sequences of putative badnaviruses and pararetroviruses, and significant differences in the prevalence of viruses between domesticated and wild *Phaseolus* spp. Our results provide insights into the distribution of viruses in domesticated and wild *Phaseolus* spp., as well as information that could be useful for the identification of emerging viruses and the development of strategies for common bean viral disease management.

## 2. Materials and Methods

### 2.1. Plant Material

Foliar tissues from 627 plants of wild (*Phaseolus* spp.) and domesticated bean (*P. vulgaris* L.) with and without symptoms of viral infection were collected in the MCD and the western State of Nayarit. The foliar tissues were collected and preserved in silica gel with a moisture indicator, as reported in Chiquito-Almanza et al. [12]. Domesticated common bean was collected from irrigated and rain-fed commercial fields in Guanajuato and Jalisco during the spring-summer growing season of 2014 and in Nayarit, Mexico, during the fall-winter growing seasons of 2013–2014 and 2014–2015. Based on their distribution and symptom diversity, including asymptomatic plants, 42 samples of wild bean and 30 samples of domesticated bean were selected for this study. The sample IDs, species, cultivar, growth stage, type of bean, collection date, and coordinates were uploaded to the National Center for Biotechnology Information (NCBI) under the BioProject number PRJNA362733; sample accession numbers are indicated in Appendix A.

### 2.2. Library Construction and Small RNA Sequencing

Total RNA was purified from 0.4 g of dried leaf tissue using TRIzol reagent following the manufacturer’s instructions (Invitrogen, Carlsbad, CA, USA); RNA quantity was determined using a NanoDrop 8000™ analyzer (Thermo Fisher Scientific, Waltham, MA, USA), and RNA quality was checked by agarose gel electrophoresis. Total RNA (3 μg) was shipped to Macrogen Inc. (Seoul, Korea), where sRNA libraries were constructed with the TruSeq^®^ Small RNA Library Preparation Kit (Illumina, San Diego, CA, USA) and sequenced on an Illumina HiSeq 2000 system in 50-bp single-end mode. Seventy-two samples were used to construct 53 sRNA libraries, 26 of which were from domesticated bean species (25 from *P. vulgaris* and one from *P. acutifolius*), while 27 were from the wild bean species *P. acutifolius*, *P. coccineus*, *P. leptostachyus*, *P. lunatus*, *P. microcarpus*, and *P. vulgaris*. All libraries were constructed from a single sample except for the composite libraries CJ16-25Lc and CJ25-56Lc from domesticated beans and SG5-136Lc, SG7-245Lc, SJ9-34Lc, SJ19-24678Lc, and SN41-2345Lc from wild beans, which were made of the equimolar mixtures of total RNA from three to six plants collected from the same spot as the single sample in the population (Appendix A).

### 2.3. Small RNA Bioinformatic Analysis

Quality check for raw sRNA reads was performed using the sRNA_clean.pl script (https://github.com/kentnf/VirusDetect/tree/master/tools/sRNA_clean, accessed on 16 October 2020) to trim the 3’ adaptor sequences and remove the sRNA reads without adaptor sequences and those of low quality or shorter than 15 nt after trimming. In order to determinate the size distribution of sRNA, trimmed sRNA sequences between 15–40 nt were aligned to the common bean genome (NCBI: GCA_000499845.1) using the Bowtie v1.1.1 program [13]; the average number of the sRNA mapped and unmapped to host were plotted. The de novo assembled of reads into contigs and identification of viruses were conducted according to Zheng et al. [14] using the online VirusDetect pipeline v1.6 with default parameters (http://virusdetect.feilab.net/cgi-bin/virusdetect/index.cgi, accessed on 18 October 2020); the common bean genome and plant virus reference database in the VirusDetect software was used as a reference to subtract host sRNA and for reference-guided assemblies through aligning sRNA sequences, and other parameters applied were coverage = 0.1 and depth = 5. SnapGene V4.1.9. (GSL Biotech LLC, Chicago, IL, USA) was used for the annotation and reconstruction of complete and near-complete genomes of viruses from the pooled sRNA reads using the complete genomes of the viruses with the highest bit score as a reference (Appendix A). The resulting viral genomes were compared to the de novo assembled contigs to ensure consistency. The percentage of coverage of the reconstructed genome upon the viral genomes used as reference was also determined. The assembled genomes were considered complete if they cover 100% of the complete genome used as reference. The distribution patterns in domesticated and wild beans from MCD and Nayarit were analyzed through a heatmap that was built using the function heatmap2 from the package gplots [15] on R 4.0.3 [16]. Code is available in Appendix A.

### 2.4. Validation of Viral Sequences by RT-PCR and Sequencing

The viruses identified by sRSA in each library were confirmed by PCR or RT-PCR in every single sample that integrated both single and composite libraries. The simultaneous extraction of both RNA and DNA viruses was performed using the cetyl trimethyl ammonium bromide (CTAB) method, according to Chiquito-Almanza et al. [12]. The absence of PCR inhibitors in total nucleic acids was tested by PCR amplification of the 26S rRNA reference gene, according to Montero-Tavera et al. [17]. Reverse transcription (RT) was conducted using 1 µg of total nucleic acids, random hexamers, and SuperScript III Reverse Transcriptase (Invitrogen, Carlsbad, CA, USA) according to the manufacturer’s instructions. The resulting cDNA was used as a template for the PCR detection of DNA and RNA viruses. Specific primers for each viral species were designed with GEMI v1.5.0 software [18] using multiple alignments of assembled contigs and complete viral genome sequences available in NCBI GenBank; the sequences were aligned using the algorithm ClustalW in BIOEDIT v7.2.5 [19]. The designed primers were analyzed with OligoAnalyzer [20], and their specificity was confirmed in silico using Primer-BLAST [21]. PCR amplifications were performed in a volume of 50 µL according to Chiquito-Almanza et al. [12]. The PCR program included one denaturation cycle at 94 °C for 3 min, followed by 35 cycles at 94 °C for 45 s, the optimal Tm of each primer pair (Appendix A) for 1 min, and 72 °C for 2 min, with a final extension step at 72 °C for 7 min. Amplicons were analyzed by 1% agarose gel electrophoresis and sequenced directly using the dideoxy terminator method. The nucleotide sequences of the PCR products were compared with BLASTn [22] to the non-redundant database to validate their sequence identity.

### 2.5. Estimation of Virus Prevalence

Virus prevalence in each region was calculated considering individual plants as the percentage of the number of individual samples with at least one virus divided by the number of total individual samples. The Mann–Whitney U-test was used to determine the significance of the differences between wild and domesticated beans concerning viral prevalence in the MCD and Nayarit using the number of samples of individual plants that contained at least one virus from the total of individual samples tested as input data.

### 2.6. Phylogenetic Analysis

The nucleotide sequences of the complete and near-complete reconstructed genomes were used to establish phylogenetic relationships with other members of their respective viral families that displayed best bit scores in BLAST searches. The phylogenetic analysis was performed using the complete genome nucleotide sequences of the isolates of *Bean common mosaic virus* (BCMV), *Bean common mosaic necrosis virus* (BCMNV), *Peanut mottle virus* (PeMoV), *Phaseolus vulgaris alphaendornavirus 1* (PvEV-1), *Phaseolus vulgaris alphaendornavirus 2* (PvEV-2), *Cowpea mild mottle virus* (CPMMV), the DNA A components of *Bean golden yellow mosaic virus* (BGYMV), and *Bean latent virus* (BLV); the amino acid sequences of the RNA-dependent RNA polymerase (RdRp) of the *Bean-associated cytorhabdovirus* (BaCV) isolate CN3; the putative novel alphaendornaviruses *Phaseolus leptostachyus alphaendornavirus* (PlepEV) and *Phaseolus lunatus alphaendornavirus* (PluEV). The RdRp domains of PlepEV and PluEV were predicted using the NCBI Conserved Domain-Search tool (https://www.ncbi.nlm.nih.gov/Structure/cdd/wrpsb.cgi, accessed on 7 December 2020). The MEGA X package [23] was used to perform the alignment of nucleotide and amino acid sequences for the selection of the best-fit substitution model for each set of data and the construction of phylogenetic trees. The nucleotide and amino acid sequences were aligned using the ClustalW algorithm with the default settings, and the phylogenetic trees were constructed using the maximum likelihood method with 1000 bootstraps. Nodes with less than 70% bootstrap values were collapsed with TreeGraph 2 ver. 2.14.0-771b [24].

## 3. Results

### 3.1. Viral Species Present in the Domesticated and Wild Bean

The sequencing of the sRNA libraries generated between 3.4 and 52.3 million raw reads per library. Trimming adapters and low-quality reads resulted in 2 to 49.2 million clean reads. In order to determine the size distribution of sRNA, trimmed reads between 15–40 nt were aligned to the common bean genome. Between 34% and 89% of the sRNA in each library were from the host; the unmapped sRNA reads with highly enriched vsiRNAs were between 11% and 66% of the total sRNA counts in each library (Appendix A). The most abundant class of unmapped to host reads was 21–24 nt in length with a peak of 21 followed by 24 nt (Figure 1).

Forty-seven complete or nearly complete genomes of nine known viral species, two sequences of 3581 bp and 2878 bp of two putative novel alphaendornaviruses, and five badnavirus- and pararetrovirus-like sequences were assembled (Appendix A). The viral species were included in five genera, where the subgroup of the (+) ssRNA viruses included the carlavirus, CPMMV, and the potyviruses BCMV, BCMNV and PeMoV. The subgroup of dsRNA viruses included the alphaendornaviruses PvEV-1 and PvEV-2, and the putative novel viruses PlepEV and PluEV; the subgroup of ssDNA included the begomoviruses BGYMV and BLV, and the (−) ssRNA subgroup included the cytorhabdovirus BaCV (Table 1). The geographic distribution of the identified viruses is shown in Figure 2. With the exception of five partial genomes of PvEV-1 and four of PvEV-2 from libraries CG3Ls, CG9Ls, CG13Ls, SG1Ls, CJ17-4Ls that were not submitted to the GenBank because their sequences showed low quality; the GenBank accession numbers of the assembled genomes and amplified fragments are indicated in Appendix A. The phylogenetic tree of all the viruses identified is shown in Appendix A.

Among the viruses identified, BaCV and BLV correspond to species recently described; thus, the characterization of their genomes was performed. The isolate of BLV identified in sample CN30 from Nayarit is a novel bipartite begomovirus whose complete genome was reconstructed and characterized by Martínez-Marrero et al. [25]. Meanwhile, the genome of the cytorhabdovirus identified in sample CN3, hereinafter referred to as BaCV isolate CN3 (BaCV-CN3; GenBank accession MT792847), consisted of 13,427 nt. It presents ORFs of the five canonical rhabdovirus structural proteins: a nucleoprotein (N) ORF1 (451 amino acids, aa), a phosphoprotein (P) ORF2 (445 aa), a matrix protein (M) ORF5 (214 aa), a glycoprotein (G) ORF6 (519 aa), and an RNA-dependent RNA polymerase (L) ORF7 (2113 aa), as well as a putative movement protein (P3) ORF3 (197 aa), and the hypothetical small protein P4 ORF4 (67 aa) (Appendix A). Except for the 3′/N region, the P3/P4 region, which does not have the conserved consensus sequences, and the region L/5′ that had 12 of the 17 conserved nucleotides, all the intergenic regions have the highly conserved sequence 3′-TAAGAAAAACYGGGAKC-5′ (Appendix A). BaCV-CN3 is closely related to the cytorhabdovirus BaCV strain BaCV-BR-GO ([26], GenBank accession MK202584], BaCV-LUZ ([27], GenBank accession MT811775), and *Papaya cytorhabdovirus* (PpVE) from Ecuador ([28], GenBank accession MH282832] (Appendix A).

Except for the sample CG32Ls of *P. acutifolius*, all the samples of domesticated beans were from *P. vulgaris*. The number of virus genera found in the MCD was lower than in the western state of Nayarit (Figure 3A). The viruses identified in the MCD were members of the potyvirus (BCMV, BCMNV, and PeMoV) and alphaendornavirus genera (PvEV-1, PvEV-2, and PlepEV), which are seed-borne except for the putative PlepEV, for which seed-borne transmission is unknown. In Nayarit, a higher diversity of viral species was identified; in addition to BCMV and BCMNV, we identified the carlavirus CPMMV, the cytorhabdovirus BaCV, the putative alphaendornavirus PluEV, and the begomoviruses BGYMV and BLV (Table 1 and Figure 3B).

### 3.2. Mixed Infections and Prevalence of Viruses in Domesticated and Wild Bean

In order to determine the presence of mixed infections, single and multiple infections were recorded. In samples from the MCD, there were 12 domesticated beans with single infections (10 with BCMV and two with PvEV-1) and eight with double infections (five with PvEV1 + PvEV-2 and three with BCMV + BCMNV). Conversely, among the wild bean samples from this region, 24 samples harbored no viruses, and five showed single infection (two with BCMV and one each with PvEV-1, PlepEV, and BCMNV) (Figure 3C). In the samples of wild bean from the MCD, BCMV was found infecting *P. vulgaris* (samples SJ9 and SG27-3) and *P. leptostachyus* (sample SJ8-5), and the latter sample was also infected with PeMoV; BCMNV was found infecting *P. leptostachyus* (sample SJ12), while PvEV-1 and PlepEV infected two samples, one of *P. vulgaris* (sample SG1-1) and one of *P. leptostachyus* (sample SG29) (Appendix A).

In the samples from domesticated beans from Nayarit, we detected single infections (2 with BCMV and one with BCMNV), double infections (one each with BCMNV + CPMMV, BCMNV + BaCV, BCMNV + BGYMV, and BCMNV + BCMV) and triple infections (one each with BCMNV + BCMV + BGYMV and BCMNV + BCMV + BLV). In the wild beans, no virus was detected in 10 samples (Figure 3C), and single infections of BCMNV and PluEV were found in two samples. The viruses identified in wild beans from Nayarit were BCMNV, infecting *P. vulgaris* (sample SN44-5), and a putative PluEV, infecting *P. lunatus* (SN35) (Appendix A).

These results show that BCMV and BCMNV were the most frequent viruses identified in domesticated and wild beans (Figure 3B and Appendix A), and single and double infections are frequent in the MCD in comparison with Nayarit, where single, double, and triple infections occur (Figure 3C). Furthermore, in all the samples of domesticated beans, at least one virus species was identified, while in wild beans, no virus was found in 34 samples (Figure 3B,C). In order to quantify these differences, the prevalence of viruses in domesticated and wild beans in each region was calculated using Mann–Whitney U-tests to compare the differences. Because all the viruses identified by sRSA were confirmed by PCR or RT-PCR in every single sample that integrated both single and composite libraries, their prevalence was calculated considering individual plants.

Twenty-one and 30 of the samples used for the construction of the 30 and 42 sRNA libraries of domesticated and wild beans, respectively, were from the MCD. The prevalence of viruses in the domesticated bean was significantly higher than in wild bean (*p* < 0.001). In the MCD, 20 of the 21 samples of domesticate bean (95.2%) were infected by at least one viral species, while in wild beans, only six of 30 samples (20%) were infected; in Nayarit, these percentages were 100% (nine of the nine samples) and 16.7% (two of the 12 samples), respectively (Figure 3D).

### 3.3. Viral Distribution Patterns in Domesticated and Wild Bean

In order to determine the distribution patterns of the viruses in domesticated and wild beans, a heatmap analysis was performed, using the coverage % of the assembled viral genome sequences upon the viral genome as reference (Figure 4). With the exception of the partial sequences of the putative novel alphaendornaviruses PluEV and PlepEV for which only a fragment was assembled, the % coverage of the assembled genomes ranged 83–100% (Figure 4, Appendix A).

The dendrogram on the left side of Figure 4 shows that grouping was based on the viruses that were found in each library, which is in accord with the domestication level (domesticated and wild). The viruses were grouped into three clusters: clusters 1 and 2 formed compact groups, while cluster 3 is integrated by two subgroups, clusters 3a and 3b (Figure 4). Clusters 1, 2, and 3a include the viruses mainly identified in libraries constructed from plant samples of domesticated beans, while in cluster 3b, with the exception of the putative alphaendornaviruses PluEV and PlepEV where no viruses were detected, these exclusively include samples of wild beans from both, the MCD and Nayarit.

Cluster 1 includes the alphaendornaviruses PvEV-1 and PvEV-2; both species were found exclusively in samples from the MCD, mainly in domesticated beans from Guanajuato (libraries CG3Ls, CG9Ls, CG13Ls, cg18Ls, CG17Ls, and CG1Ls) and Jalisco (library CJ14-4Ls); meanwhile, PvEV-1 was also identified in one sample of wild bean from Guanajuato (library SG1Ls).

Cluster 2 encompasses the higher diversity of viral species. Among the six viruses in this cluster, BCMNV is present in all the libraries and includes mainly samples of domesticated beans from Nayarit, and two samples of wild beans, one from Nayarit (SN44-5Ls) and one from Jalisco (SJ12Ls). This result confirms the high prevalence of BCMNV in common beans grown in Nayarit and the presence of this virus in wild and domesticated bean from Nayarit and Jalisco (library CJ25-56Lc in cluster 3a).

Cluster 3a includes all libraries infected with BCMV, except those from libraries CN61Ls and CN30Ls from cluster 2. In addition, PeMov was identified in mixed infection along with BCMV in library SJ8-5Ls. Although most of the libraries in this cluster are from samples of domesticated beans, it also includes samples of wild beans (SJ9-34Lc and SG27-3Ls). Samples in this cluster were collected from Nayarit, Guanajuato and Jalisco. This result indicates the wide distribution of BCMV in the common bean that is grown in this region and confirms the presence of this virus in wild beans in the MCD.

### 3.4. Identification of Novel Viral Species and Viral-Like Sequences

Several assembled contigs without detectable nucleotide homology but with amino acid identity to reference virus protein sequences were identified in 11 of the libraries (Appendix A). In order to determine if these sequences corresponded to novel viral species, the assembled contigs were used as references to design specific primers. Through standard DNA amplification techniques and sequencing, the partial nucleotide sequences of two putative novel alphaendornaviruses and five virus-like sequences were reconstructed. Below we describe the characteristics of the reconstructed sequences of both putative novel alphaendornaviruses and the five virus-like sequences.

#### 3.4.1. Identification of Two Putative Alphaendornaviruses

The partial nucleotide sequences of two putative novel alphaendornaviruses were reconstructed from the wild asymptomatic samples SG29 from *P. leptostachyus* collected in Guanajuato and SN35 from *P. lunatus* collected in Nayarit. For identification purposes, the names *Phaseolus leptostachyus alphaendornavirus* (PlepEV) and *Phaseolus lunatus alphaendornavirus* (PluEV) were assigned. The partial nucleotide sequence of PlepEV consisted of 3581 nt (GenBank accession MT792848) and included one partial ORF of 3540 nt that was translated into a fragment of a putative polyprotein of 1179 aa, sharing 45.9% and 43.9% aa identity with the carboxy-terminal polyprotein of the unclassified alphaendornavirus *Geranium carolinianum endornavirus* (GcEV; unpublished; GenBank accession MH577297) and the *Phaseolus vulgaris endornavirus* 3 isolate LA (PvEV-3; [29]; GenBank accession MG242064), respectively. The query coverage of both aa sequences was 96%. Furthermore, the partial nucleotide sequence of PluEV consisted of 2878 nt (GenBank accession MT792849), which includes one partial ORF of 2874 nt that was translated into a putative polyprotein fragment of 958 aa, sharing 65.1% aa identity (98% query cover) with the carboxy-terminal polyprotein of the PvEV-3 isolate LA and 46% aa identity (97% query cover) with the carboxy-terminal polyprotein of GcEV.

The aa identity between the predicted polyproteins of PlepEV and PluEV was 44.9%. The nucleotide sequences of the fragments amplified by PCR with specific primers (Appendix A) for PlepEV (1076 bp) and PluEV (1090 bp) exhibited 100% identity with the respective reconstructed sequences. The predicted amino acid sequences of both reconstructed sequences of PlepEV and PluEV included the full RdRp domain. The phylogenetic analysis of this domain with different alpha- and betaendornaviruses show that PlepEV and PluEV clustered together with PvEV-3 and GcEV in the clade corresponding to the alphaendornavirus genus (Figure 5).

#### 3.4.2. Identification of Virus-Like Sequences

Five virus-like sequences with amino acid identity to the viral reference sequences were reconstructed (Appendix A). For identification purposes, the sequences were named based on the sample from where they were obtained and the genus of the virus to which they match aa identity, as *Phaseolus vulgaris badnavirus*-like sequence CG17 (PvBV-CG17), *Phaseolus coccineus badnavirus*-like sequence SJ10-3 (PcBV-SJ10-3), *Phaseolus coccineus pararetrovirus*-like sequence SJ10-3 (PcPRV-SJ10-3), *Phaseolus lunatus badnavirus*-like sequence SN35 (PluBV-SN35), and *Phaseolus lunatus pararetrovirus*-like sequence SN35 (PluPRV-SN35).

The sequence of PvBV-CG17 consisted of 1629 bp and was reconstructed from a sample from common bean cv. Azufrado Higuera with symptoms of local chlorosis lesions; in this sample, PvEV-1 (GenBank accession MG640416) was also identified. The sequence of PcBV-SJ10-3, consisting of 2604 bp, was reconstructed from a wild *P. coccineus* sample with symptoms of local chlorosis lesions, while the sequence PluBV-SN35, consisting of 1378 bp, was reconstructed from a symptomless wild *P. lunatus* sample; in both samples, SJ10-3 and SN35, the pararetrovirus-like sequences PcPRV-SJ10-3 (2588 bp) and PluPRV-SN35 (1440 bp) were also identified; these translated into 862 aa and 480 aa proteins, respectively, with 49% and 32% aa identity to *Citrus endogenous pararetrovirus* polyprotein (GenBank accession KF800044).

Both the PvBV-CG17 and PluBV-SN35 sequences exhibited one partial ORF, which was translated into putative proteins of 543 and 456 aa, respectively, with 39.0% and 35.6% aa identity to the putative P3 polyprotein (ORF3) of the unclassified *Badnavirus-like pelargonium vein banding virus* (PVBV; GenBank accessionGQ428155). On the other hand, the sequence of PcBV-SJ10-3 exhibited two partial ORFs, one was translated into a putative protein of 58 aa with 41.3% aa identity to the ORF2 protein of the badnavirus cacao swollen shoot virus (CSSV; GenBank accession MF642734), and the other was translated into a putative protein of 808 aa with 43.4% aa identity to the P3 putative polyprotein of PVBV. Although the reconstructed sequences of PvBV-CG17, PcBV-SJ10-3, and PluBV-SN35 shared low nucleotide similarity with the available sequences of badnavirus in the GenBank, suggesting that they could correspond to a novel badnavirus species, the reconstructed fragment did not include the RT/RNAse H domain required for the demarcation of the species. Additionally, since badnaviruses and pararetroviruses had been reported as integrated fragmented sequences, the circularity of the genome needs to be confirmed.

## 4. Discussion

### 4.1. Identification and Distribution of Viruses in Domesticated and Wild Bean

In addition to the MCD, samples of wild and domesticated *Phaseolus* spp. were collected in the western state of Nayarit since the occurrence of virus damaging in the common bean crop is particularly high in this area [12,30], and it is located near the MCD. The collection of samples at a site was based on the presence of viral symptoms, and symptomless plants were also sampled. However, in most of the samples of wild *Phaseolus* spp. with presumptive viral symptoms, no viruses were identified; the symptoms observed in those samples could be due to nutrient deficiencies or the effect of other biotic or abiotic stresses that may show similarities to mosaic or mottling [31]. Nine species were identified in the MCD and Nayarit, with a higher diversity of viral species in Nayarit, which includes six species (BCMV, BCMNV, BGYMV, BLV, CPMMV, and BaCV). BCMNV was mainly found in Nayarit, and BCMV was widely distributed in both the MCD and Nayarit. PeMoV was identified in one sample from Jalisco; PvEV1 and PvEV-2 were found exclusively in Guanajuato (Table 1, Figure 2 and Figure 4).

The number of virus genera found in the MCD was lower than in the western state of Nayarit (Figure 3A); a possible explanation for the higher incidence of viruses in Nayarit could be that as an important bean producing area during the fall-winter cycle, Nayarit was one of the chosen locations for the establishment of international common bean trials in Mexico, which facilitated the introduction of the seed-borne viruses [32,33,34]. Additionally, the abundant presence of the whitefly *B. tabaci* represents a constraint to vegetable production [35]. Moreover, the detection of a high proportion of mixed infections in Nayarit suggests that the climatic conditions and cultural practices, such as the exchange of seeds between growers or the use of the grain of susceptible cultivars as seed, favor the dissemination of viruses that is further enhanced by the presence of the vector.

### 4.2. Prevalence of Viruses in Domesticated and Wild Bean

The evidence indicates that the incidence of viral diseases in the wild populations sampled is low in contrast to domesticated bean, in which almost all samples showed single, double, or triple infections (Figure 3B,C). The higher prevalence of viruses in the domesticated bean (Figure 3D) could be the result of anthropogenic influence. Cultivation is a major cause of environmental heterogeneity [36], and monoculture provides suitable conditions for the fast emergence and spread of plant viruses and their vectors [1], which is a situation that does not occur in wild bean populations that are in distinct ecological conditions. Studies of virus infection in chiltepin populations (*Capsicum annuum* var. *glabriusculum*), a wild pepper plant under different anthropogenic influences, including cultivation and incipient domestication, support the fact that ecological conditions are a valuable predictor of virus prevalence and that human management is associated with a reduction of habitat biodiversity and genetic diversity of host species; in addition, as host plant density is increased, the viral disease risk is enhanced due to plant-to-plant proximity. Furthermore, virus infection decreases the survival and fecundity of wild chiltepin populations, hence negatively affecting fitness [36]. It is also likely that the variation of traits, such as resistance to viruses and other quantitative traits found in wild beans, are not present in the domesticated common bean due to genetic diversity bottlenecks during bean domestication [37,38]. Nonetheless, the rate of the discovery of putative novel viruses and virus-like sequences in wild beans was high in comparison to the domesticated bean, in which almost all the species identified were already known. These results agree with those of Bernardo et al. [39], who found that prevalence and identified family-level virus diversity were greatest in cultivated areas, and unknown virus species were found primarily from uncultivated plants. Ultimately, the number of studies on wild bean populations, as compared to the domesticated form, is low.

### 4.3. Characteristics of Known Viruses Identified

The viruses BCMV, BCMNV, BGYMV, and CPMMV have been documented infecting common bean and wild bean in Guanajuato, Jalisco, and Nayarit, Mexico [12,30,40]. In contrast, the viruses PvEV-1, PvEV-2, PeMoV, and BaCV have not been reported infecting *Phaseolus* spp. in Mexico before.

Predicting epidemics of plant virus diseases and the causes that facilitate pathogen emergence is complex and depends on multiple factors, reviewed in [1,41,42]. Besides these factors, the spread from the main reservoir into a new environment, the establishment of infection, and effective mechanisms of transmission between hosts are steps required for emergence to occur [43]. In this context, the virulence and the mechanism of transmission of a virus, particularly its capacity for seed-borne and insect transmission, are some of the features intrinsic to the virus that could contribute to define its potential to become an emerging pathogen. BGYMV, BaCV, and CPMMV are transmitted by *B. tabaci*. BGYMV is an important non-seed-transmitted virus in areas with a high incidence of *B. tabaci*; the disease incidence and the extent of losses vary depending on the populations of the whitefly, the susceptibility of the bean cultivar, and environmental conditions, particularly rainfall, which affects both *B. tabaci* populations and cultural practices [44]. The vector that spreads BLV and the affectation that may cause to the common bean production are unknown; under experimental conditions, this begomovirus does not cause symptoms in common bean [25].

Based on the high nucleotide sequence identity between the cytorhabdovirus BaCV-BR-GO and PpVE, it was proposed that the virus species would be named *Papaya cytorhabdovirus* with strains PpVE infecting papayas and BaCV infecting beans [45]; thus, BaCV-CN3 is likely a strain of BaCV since it infects common bean and is closely related to the strains BaCV-BR-GO, BaCV-LUZ, as shown in the phylogenetic tree based on the RdRp amino acid sequences (Appendix A). The vector of PpVE is unknown, and although BaCV is the first rhabdovirus reported to be transmitted to common beans by whitefly *B. tabaci* MEAM1 [27], the transmissibility of BaCV-CN3 by this vector was not determined. The symptoms induced by BaCV-CN3 could neither be confirmed because the sample from which it was identified was co-infected with BCMNV (GenBank accession MK069983). Interestingly, BaCV and PpVE have also been identified in mixed infections. Pinheiro-Lima et al. [27] found that 85 of the 91 common bean samples that tested positive to BaCV were co-infected with CPMMV, BGMV, or with CPMMV and BGMV, and although six samples were “single infected,” it is also possible that these samples were infected with another virus distinct to the ones tested; similarly, all the plants that tested positive to PpVE were also positive for *Papaya ringspot virus* (PRSV), and no differences in leaf symptoms were observed between PRSV single-infected plants and plants co-infected with both viruses [28]. Therefore, it is important to investigate if this group of viruses requires a helper virus, are seed-borne transmitted, and cause important economic losses.

The presence of endornaviruses PvEV-1 and PvEV-2 in almost all the domesticated and wild samples from Guanajuato (Figure 4) could be related to the domestication process of the common bean, which likely occurred in the Lerma-Santiago basin in the states of Jalisco and Guanajuato, Mexico [7]. Endornaviruses are persistent dsRNA viruses that are transmitted vertically [46] and develop a highly symbiotic relationship with the host plant [47]. PvEV-1 and PvEV-2 are transmitted to the progeny of *P. vulgaris* plants at rates close to 100% [48] and have been detected in many genotypes of Mesoamerican origin but rarely in genotypes of the Andean pool, suggesting that these two endornaviruses likely originated from infected wild species that were selected during common bean domestication and introduced into some cultivars during the breeding process [49].

The main host of PeMoV is peanuts (*Arachis hypogea*), but it also infects susceptible genotypes of *P. vulgaris*, causing various reactions that range from mosaic, banding, leaf deformation and blistering, to systemic necrosis in genotypes that have only the dominant *I* gene; these reactions are similar to those caused by necrotic strains of BCMV and BCMNV. Although PeMoV is mechanically and non-persistently transmitted by aphids, transmission to common bean seeds is low (1%) [50].

Among the viruses identified in this study, BCMV and BCMNV likely present the highest phytosanitary risk due to their high incidence in the region and their efficient mechanisms of seed and vector transmission [51]. BCMV and BCMNV are considered one of the major production constraints of common bean in Latin America, the Caribbean, and East, West, and Southern Africa and the most worldwide spread viruses [52,53]; the percentage of seed transmission of these viruses is as high as 50% [51] and hence are readily spread within and between countries [50,54]. BCMV can remain viable in bean seeds for more than 30 years [54], and under field conditions, the infected seedlings originated from BCMV- or BCMNV-infected seeds act as primary foci, and subsequently, the virus spreads through several aphid species in a non-persistent manner [55]. Yield losses of 53–83% in beans have been attributed to BCMV [56], and even symptomless BCMV infections can induce yield losses higher than 50% in susceptible common bean genotypes [2]. The high incidence of BCMV and BCMNV in wild *Phaseolus* spp. from the MCD and the western state of Nayarit suggest that the transmission of these viruses through insect vectors is highly efficient. Further studies are needed to know how and from where these viruses came into the wild bean populations of the MCD and Nayarit and if transmission occurred from cultivated into wild bean populations. The other virus that has the potential to emerge as an important pathogen is CPMMV, which is emerging as a soybean pathogen in Brazil, and it is re-emerging in cultivated common beans [57], particularly in genetically modified common bean cultivars resistant to BGMV [58]. CPMMV is the only *Carlavirus* that is not transmitted by aphids, and it is nonpersistently transmitted by *B. tabaci* (MEAM1), with a period of access to acquisition of 10 min and an inoculation period of 5 min [59]. Although the seed transmission of CPMMV has been reported, the different rates of seed transmission in cowpea, soybean, and common bean suggest that this characteristic depends on the isolate, the strain, and the cultivar of the host, which can make the control of the disease difficult and facilitate its spread due to the existence of asymptomatic infections [50,58]. Furthermore, symptoms are rather variable, such that some infections of CPMMV have been confounded as new and different viral infections [58,60]. In soybean, CPMMV causes stem necrosis disease, but distinct symptoms have been observed in the same cultivar within the same cultivation field [58]. In common beans, CPMMV causes angular mosaic disease, including symptoms of systemic mosaic and leaf distortion, although the infection may be asymptomatic depending on the cultivar [40,50]. In Mexico, this virus has been identified in infections of common bean (*P. vulgaris*) and wild bean (*P. vulgaris*, *P. leptostachyus*, and *P. coccineus*), and some RT-PCR-positive samples did not present symptoms [40].

CPMMV is classified in Annex 1/A1 in the European Union and in the United States as a quarantine pest, where it is considered to represent a potential risk to soybeans, beans, cowpea, and some species of the Solanaceae family [61,62]. Mexico does not regulate CPMMV, but its occurrence in beans could have negative consequences for sales abroad. To determine the phytosanitary risk of CPMMV, it is necessary to understand its viral genomic diversity, host range, capacity for seed-borne transmission, distribution, prevalence, and incidence, especially in areas with high whitefly vector pressure and where common beans are grown from seed or grain produced in situ [40].

### 4.4. Putative Novel Viruses Identified

The assembled sequences of PlepEV and PluEV share amino acid identity with the RdRp domains of PvEV-3 and GcEV, suggesting that they could be novel alphaendornaviruses; however, their complete genome sequences will be required for their classification. The plant samples in which PlepEV and PluEV were identified were symptomless, and like other endornaviruses that have been reported in economically important crops, their effect on the host has not been fully determined. For instance, PvEV-1 and PvEV-2 were not associated with visible pathogenic effects on *P. vulgaris* cv Black Turtle Soup [63]; also, the interaction of these viruses with other common bean viruses and pathogens should be tested for possible interactions.

The badnavirus- and pararetrovirus-like sequences reconstructed in this study were obtained by sequencing since multiple single-nucleotide variations (SNVs) in the assembled contigs made scaffolding difficult. Some badnaviruses occur as integrated, complete, fragmented, and/or rearranged genomic sequences of the original circular viral DNA in their host plant genomes, in which case they are referred to as endogenous pararetroviruses or endogenous viral elements [64]. Their integration takes place by illegitimate recombination into host genomes, and their presence is not necessarily associated with infection. The identification of putative badnaviruses or pararetroviruses highlights the need for further specific studies to determine their biology and phytosanitary risks.

## 5. Conclusions

Higher viral diversity was found in Nayarit in comparison with the MCD. BCMV and BCMNV were the most prevalent viruses in wild and domesticated beans; these viruses, along with CPMMV, have the potential to emerge as an important pathogen because they are seed-borne and non-persistently transmitted viruses. Integrated disease management approaches using virus-resistant common bean cultivars and efficient management of insect vectors and weeds can reduce the dispersion and the production losses associated with these viruses. More studies are needed to confirm the putative viruses identified in this study and determine the effect of human management on population heterogeneity, biodiversity loss, and disease risk that these viruses represent for the domesticated common bean and wild bean populations.

## Figures and Tables

**Figure 1 viruses-13-01153-f001:**
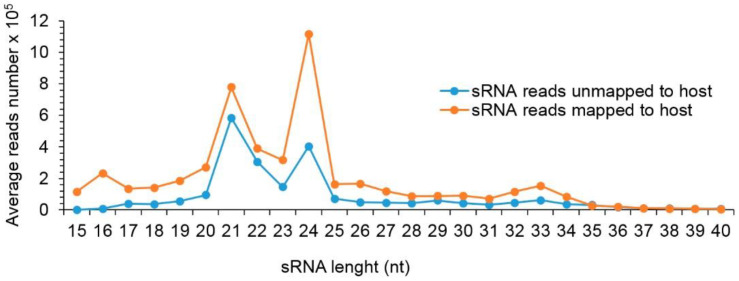
Small RNA size distribution of reads mapped and unmapped to host. Average number of reads of mapped and unmapped to host are shown in orange and blue lines, respectively.

**Figure 2 viruses-13-01153-f002:**
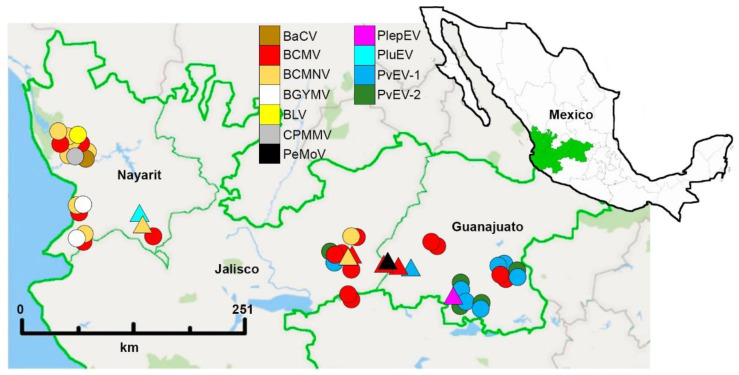
Virus distribution infecting *Phaseolus* spp. in the Mesoamerican center of domestication of common bean (Jalisco and Guanajuato States) and Nayarit, Mexico. The colors indicate the corresponding viral species. Circles and triangles correspond to the viruses identified in domesticated and wild *Phaseolus* spp., respectively. Geographic coordinates are indicated in Appendix A.

**Figure 3 viruses-13-01153-f003:**
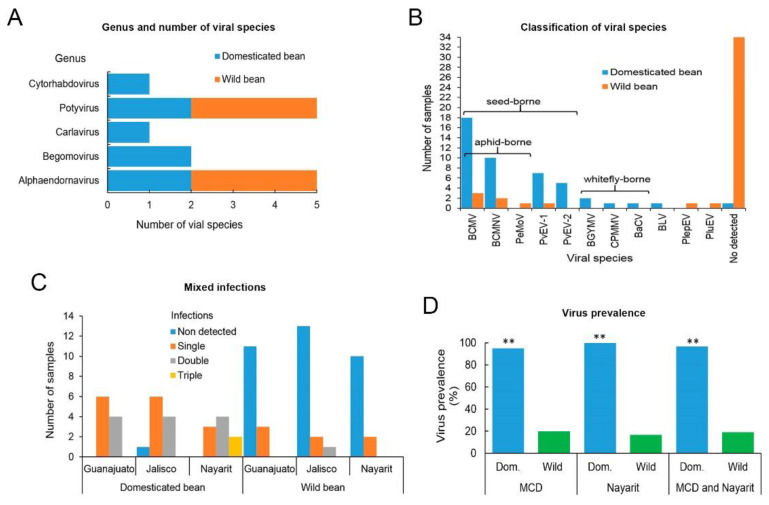
Virus detection, mixed infections, and prevalence of viruses in domesticated and wild *Phaseolus* spp. samples from the Mesoamerican center of domestication of common bean (Jalisco and Guanajuato States) and Nayarit, Mexico. (**A**) Genus and number of viral species detected in domesticated and wild bean in the MCD and Nayarit. (**B**) The classification of viral species detected by their mechanism of transmission, number of samples, and host (domesticated and wild bean). (**C**) A summary of samples organized by the number of detected viruses per sample and region. (**D**) Virus prevalence associated with domesticated (Dom.) and wild bean (Wild) in the MCD and Nayarit. Significant differences in virus prevalence are indicated by ** = *p*-value < 0.01.

**Figure 4 viruses-13-01153-f004:**
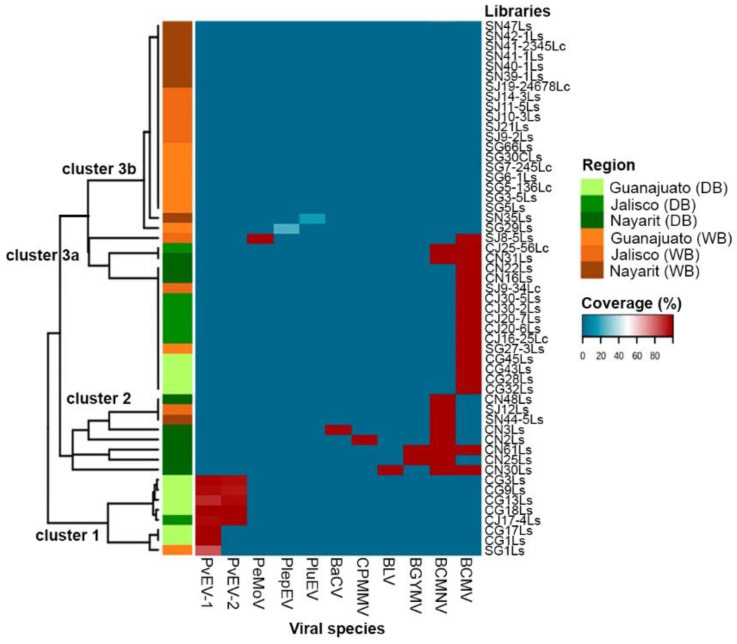
Viral distribution patterns in domesticated and wild bean samples from the Mesoamerican center of domestication of common beans (Guanajuato and Jalisco States), and Nayarit, Mexico. On the left side, the infection profiles are clustered and colored according to the region of provenance. Coverage % of reconstructed genome upon genomes used as reference is shown. DB: domesticated bean; WB: wild bean.

**Figure 5 viruses-13-01153-f005:**
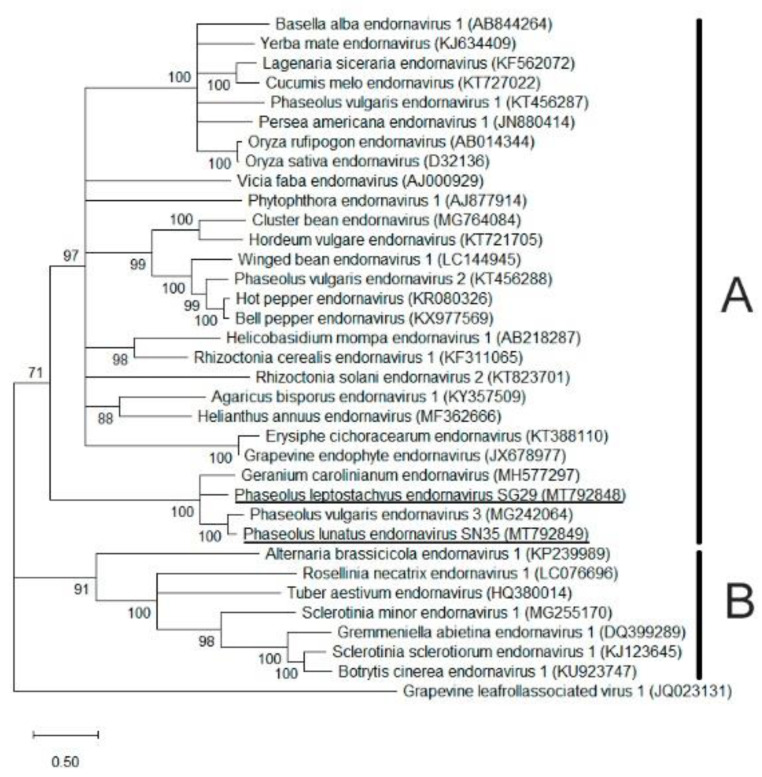
Phylogenetic tree of putative new alphaendornavirus PlepEV and PluEV: (**A**) Alphaendornavirus clade and (**B**) Betaendornavirus clade. Phylogenetic trees were constructed using an alignment of the amino acid sequences of the RdRp domains with the Maximum-Likelihood (LG + G + I model) with 1000 bootstraps. Grapevine leafroll-associated virus 1 (genus Ampelovirus, family Closteroviridae) was included as an outgroup. Nodes with less than 70% bootstrap values are collapsed. PlepEV and PluEV are underlined.

**Table 1 viruses-13-01153-t001:** Distribution, numbers of assembled genomes, and transmission of viruses infecting *Phaseolus* spp. in the MCD and Nayarit, Mexico.

Virus Name	Acronym	Genus	Family	Genome *	MCD		Nay.	Genomes Assembled ^§^	Transmission	
					Gto.	Jal.			Seed	Vectors
*Phaseolus vulgaris alphaendornavirus 1*	PvEV-1	*Alphaendornavirus*	Endornaviridae	dsRNA	+	+		8	+	U
*Phaseolus vulgaris alphaendornavirus 2*	PvEV-2	*Alphaendornavirus*	Endornaviridae	dsRNA	+	+		5	+	U
*Bean common mosaic virus*	BCMV	*Potyvirus*	Potyviridae	ssRNA (+)	+	+	+	18	+	Aphids
*Bean common mosaic necrosis virus*	BCMNV	*Potyvirus*	Potyviridae	ssRNA (+)		+	+	10	+	Aphids
*Peanut mottle virus*	PeMoV	*Potyvirus*	Potyviridae	ssRNA (+)		+		1	+	Aphids
*Cowpea mild mottle virus*	CPMMV	*Carlavirus*	Betaflexiviridae	ssRNA (+)			+	1	+	*B. tabaci*
*Bean-associated cytorhabdovirus*	BaCV	*Cytorhabdovirus*	Rhabdoviridae	ssRNA (-)			+	1	U	*B. tabaci*
*Bean golden yellow mosaic virus* DNA A	BGYMV DNA A	*Begomovirus*	Geminiviridae	ssDNA			+	2	−	*B. tabaci*
*Bean golden yellow mosaic virus* DNA B	BGYMV DNA B	*Begomovirus*	Geminiviridae	ssDNA			+	2	−	*B. tabaci*
*Bean latent virus* DNA A	BLV DNA A	*Begomovirus*	Geminiviridae	ssDNA			+	1	U	U
*Bean latent virus* DNA B	BLV DNA B	*Begomovirus*	Geminiviridae	ssDNA			+	1	U	U
*Phaseolus leptostachyus alphaendornavirus*	PlepEV	Putative *Alphaendornavirus* ^¶^	Endornaviridae	dsRNA	+				U	U
*Phaseolus lunatus alphaendornavirus*	PluEV	Putative *Alphaendornavirus* ^¶^	Endornaviridae	dsRNA			+		U	U
Badnavirus-like sequence	PvBV-CG17	Putative *Badnavirus* ^¥^	Caulimoviridae	dsDNA	+				U	U
Badnavirus-like sequence	PcBV-SJ10-3	Putative *Badnavirus* ^¥^	Caulimoviridae	dsDNA		+			U	U
Badnavirus-like sequence	PluBV-SN35	Putative *Badnavirus* ^¥^	Caulimoviridae	dsDNA			+		U	U
Pararetrovirus-like sequence	PcPRV-SJ10-3	Not assigned ^¥^	Caulimoviridae	dsDNA		+			U	U
Pararetrovirus-like sequence	PluPRV-SN35	Not assigned ^¥^	Caulimoviridae	dsDNA			+		U	U

^¶^: Putative 1. ^¥^: Putative 2. * Baltimore class d (W) Domestic. ^§^: Number of complete or near-complete genomes assembled. MCD: Mesoamerican Center of Domestication. Gto.: Guanajuato; Jal.: Jalisco; Nay.: Nayarit. U: Unknown.

## Data Availability

The sRNA reads were uploaded to the NCBI Sequence Read Archive (SRA) under accession numbers SRX3217971-SRX3218007. The sequences of the viral genomes reconstructed, as well as the PCR-amplified fragments for the validation of the viruses detected, were uploaded to the GenBank, and the accession numbers are indicated in Appendix A.

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
