# Peer review of "Diversity and Distribution of Viruses Infecting Wild and Domesticated Phaseolus spp. in the Mesoamerican Center of Domestication"

_viruses, 2021, doi:10.3390/v13061153_

Round 1

Reviewer 1 Report

Viruses 2021 Review

Title: Diversity and distribution of viruses infecting wild and domesticated Phaseolus spp., in the Mesoamerican center of domestication

Summary:

The authors identify viruses and their distribution in infected Phaseolus spp.at the center of domestication and biodiversity in Mexico.  This work is important because common bean (Phaseolus vulgaris L.) is highly virus-susceptible and the most widely cultivated member of the Leguminosae worldwide. The work addresses the impact and threat of virus disease to Phaseolus biodiversity, a critical element of breeding and disease control strategies at centers of origin and domestication. The study describes the identification and distribution of viruses infecting wlld and domesticared Phaseolus spp. at the center of origin and domestication in the Mesoamerican states of Jalisco, Guanajuato (MCD) as well as Nayarit, a state adjacent to the MCD. The information described in this work can be used to monitor and potentially control emerging indigenous and newly introduced viruses that potentially threaten the common bean at the biodiversity center.

The threat of virus infection on Phaseolus biodiversity at the MCD is the result of combined influence of several factors, including current and emerging viruses, modes of virus spread, the status of viral host defense in different Phaseolus species and virus suppression of host defense. It will be interesting to use this and ongoing work to identify continuing and emerging new virus diseases and use this system to add to understanding to plant defense and virus counter defense mechanisms.

General comments. The paper is well-written and thorough. In general the data are presented clearly, are well-documented and understandable. The results of the authors’ quantitative analyses to identify and determine the distribution of viruses in the MCD and Nayarit address the major goal of the paper: to identify emerging viruses and their distribution to develop strategies for viral disease management of bean.

Revisions. A few revisions in the writing are suggested to highlight important results and appeal to a broader audience.

Abstract:

  1. Consider reorganizing abstract to introduce the problem addressed by the study first. Some detail can be removed. See revised Suggested Abstract below. The original abstract is also shown.

Suggested Abstract: To evaluate the impact of virus disease on Phaseolus biodiversity we determined the identity and distribution of viruses infecting wild and domesticated Phaseolus spp. in the Mesoamerican Center of Domestication (MCD) and the western state of Nayarit in Mexico. We used small RNA sequencing and assembly to identity complete or near complete sequences of forty-seven genomes belonging to nine viral species of five viral genera, as well as partial sequences of two putative new endornaviruses and five badnavirus- and pararetrovirus-like sequences.  The prevalence of viruses in domesticated bean was significantly higher (how much higher?) than the prevalence in wild bean (p<0.001) and all domesticated bean samples were positive for at least one virus. In contrast, no viruses were detected in 80-83% of the wild bean samples tested. The Bean common mosaic virus and Bean common mosaic necrosis potyviruses were the most prevalent viruses in wild and domesticated beans, while Cowpea mild mottle virus, transmitted by the whitefly Bemisia tabaci, has the potential to emerge as an important pathogen because it is both seed-borne and a non-persistently transmitted virus. These results provide insights into the distribution of viruses in cultivated and wild Phaseolus spp. and will be useful for the identification of emerging viruses and the development of strategies for common bean viral disease management.

Current Abstract: The diversity and distribution of viruses infecting wild and domesticated Phaseolus spp., in the Mesoamerican center of domestication (MCD) and the western state of Nayarit, Mexico, were  investigated using small RNA sequencing technology and validated with targeted nucleic acid amplification and dideoxy terminator sequencing. The species included were P. vulgaris, P. acutifolius P. leptostachyus, and P. lunatus. Forty-seven complete or nearly complete genomes of nine viral species ranged from five genera, two partial sequences of two putative new species of alphaendornavirus, and five virus-like sequences of putative badnaviruses and pararetroviruses were assembled. The prevalence of viruses in the domesticated bean was significantly higher than the prevalence in wild beans (p<0.001). No viruses were detected in 80-83% of the samples from wild bean plants from  the MCD and Nayarit, in contrast, all samples of domesticated bean were positive for at least one virus. Bean common mosaic virus and Bean common mosaic necrosis virus were the most prevalent species in wild and domesticated beans, while Cowpea mild mottle virus has the potential to emerge as an important pathogen because is seed-borne and non-persistently transmitted by Bemisia tabaci.  These results provide insights into the distribution of viruses in cultivated and wild Phaseolus spp. and could be useful for the identification of emerging viruses and the development of strategies for common bean viral disease management.

  1. Introduction
  2. The introduction is good. I suggest separating the paragraphs starting at line 45, then 53 and then 66.
  3. Rephrase to clarify the meaning of “human management of disease risk” and how it is or isn’t connected to “the impact of viruses on bean cultivation.” Line 343
  4. Results
  5. To reach a broader audience begin Results sections 3.1, 3.2, 3.2.1, and 3.2.2 by stating the problem or question addressed in each and at the end of each section state a conclusion or the main finding of that section. (e.g., to begin “In order to…” and at the end “These results show….”.) Begin each with the question or the problem addressed and finish each section with a very brief conclusion, main finding, or takeaway of the results described in each.
  6. Move the end paragraph of 3.1 to the beginning of 3.1 to lead off the section .
  7. 3.1. Include a new Table 1 listing virus names, acronyms, taxonomy, wild or domestic host species, and geographical distribution (see the example Suggested Table 1) as a convenient reference to the important results in 3.1. I suggest that Table 1(see the example Suggested Table 1) be displayed on the same page as a new map listed as Figure 1 (currently Figure 1A.). In the new Figure 1 map, distinguish wild and domestic virus hosts using different colors. Co-located W and D spots can be indicated as divided circles using two colors side by side in the circle. Table 1 is more easily accessed than the complete Table S1 and provides the important details described in section 3.1 and can be referred to throughout the in the paper. The  table could be shown together with just the map in Figure 1A before figure 1 of the current paper.
  8. 3.1. Figure 1B, now Figure 2. Comparison of the size distribution of sRNAs mapping to virus or host genomes is interesting. Move Figure 1B to supplementary figures and include a description of the approach and analyses of results in Figure 2B.
  9. 3.1. In former Figure 1B, now Figure 2, I suggest using line graphs to display an average of sRNA sizes in libraries to compare sRNA size distribution in the host compared to the virus.
  10. 3.1. In former Figure 1, now Figure 2, a short subtitle above each graph would be helpful to determine the results shown in each.
  11. 7. 3.2 To better understand the graphical depiction of results in Figure 2, now Figure 3, I suggest dividing Figure 2 into 2 parts, 2A. The figure title should reveal the main finding of the figure. Divide top and bottom panels into A and B and turn each graphic 90º counterclockwise. Describe the clustering A.

Figure 2 is not described in text. In the text provide a clear description of Figure 2 (Figure 3A and 3B), including methods and results. Include the results and conclusions of the clustering shown in the top panel. What is the relationship between the clustering and the regional distribution shown in the lower section? Please include a description of the conclusion based on the results shown in lower part of the graph. The font used for the library names in the X-axis is too small.

Discussion

  1. Add subtitles to separate sets of paragraphs in the discussion to identify sections according to the main issues and conclusions addressed.

Author Response

Thank you very much for the comments and suggestions, we are grateful for your help.

  1. Consider reorganizing abstract to introduce the problem addressed by the study first. Some detail can be removed. See revised Suggested Abstract below. The original abstract is also shown.

Response: Original abstract was replaced by the suggested abstract by the reviewer with minor modifications. The value of the prevalence of viruses in domesticated bean vs wild bean (97 vs 19%; p>0.001) was included.

  1. Introduction
  2. The introduction is good. I suggest separating the paragraphs starting at line 45, then 53 and then 66.

Response: The paragraphs were separated as suggested by the reviewer.

  1. Rephrase to clarify the meaning of “human management of disease risk” and how it is or isn’t connected to “the impact of viruses on bean cultivation.” Line 343

Response: Paragraph in Line 343 was changed as follows: “Understanding the diversity and evolution of the prevalent viruses in wild and domesticated Phaseolus spp. is needed to quantify the impact of viruses on bean cultivation and the effect of human management on population heterogeneity, biodiversity loss and disease risk.”

  1. Results
  2. To reach a broader audience begin Results sections 3.1, 3.2, 3.2.1, and 3.2.2 by stating the problem or question addressed in each and at the end of each section state a conclusion or the main finding of that section. (e.g., to begin “In order to…” and at the end “These results show….”.) Begin each with the question or the problem addressed and finish each section with a very brief conclusion, main finding, or takeaway of the results described in each.

Response: The suggested recommendation was done in most of the sections.

  1. Move the end paragraph of 3.1 to the beginning of 3.1 to lead off the section.

Response: The end paragraph of section 3.1 was moved to the beginning of this section.

  1. 3.1. Include a new Table 1 listing virus names, acronyms, taxonomy, wild or domestic host species, and geographical distribution (see the example Suggested Table 1) as a convenient reference to the important results in 3.1. I suggest that Table 1(see the example Suggested Table 1) be displayed on the same page as a new map listed as Figure 1 (currently Figure 1A.).

Response: Table 1 was included with all the information suggested by the reviewer with minor modifications. Additional information according to the suggestion of reviewer 2 was also included. Table 1 contains the virus names, acronym, genus, family, genome type, total number of complete or near complete sequences of assembled genomes, and the transmission of identified viruses was included. The Title was modified according to its content.

In the new Figure 1 map, distinguish wild and domestic virus hosts using different colors. Collocated W and D spots can be indicated as divided circles using two colors side by side in the circle. Table 1 is more easily accessed than the complete Table S1 and provides the important details described in section 3.1 and can be referred to throughout the in the paper. The table could be shown together with just the map in Figure 1A before figure 1 of the current paper.

Response: A map with the viruses found in wild and domesticated samples was included as Figure 2 in the present version of the manuscript. Different colors were used to distinguish the viral species; circles represent the viruses found in samples of domesticate bean and triangles the viruses found in wild bean. Table 1 is included before Figure 2. 

  1. 3.1. Figure 1B, now Figure 2. Comparison of the size distribution of sRNAs mapping to virus or host genomes is interesting. Move Figure 1B to supplementary figures and include a description of the approach and analyses of results in Figure 2B.

Response: Since our goal was to identify the viruses prevalent in the MCD and in Nayarit, and as the software VirusDetect performs de novo assembly of contigs and virus identification using cleaned reads as input, we only performed the subtraction of sRNAs derived from the host genome as an approximation to determine the size distribution of unmapped sRNAs, which are highly enriched with vsiRNAs. The alignment of the unmapped-host reads with viral genomes could have been performed to determine the size distribution of sRNA for viruses, but we did not do it, nevertheless, the lack of this result does not affect the identification of viruses.

In Materials and methods section 2.3 the approach was included: “In order to determine the size distribution of sRNA, trimmed sRNA sequences between 15-40 nt were aligned to the common bean genome (NCBI: GCA_000499845.1) using the Bowtie v1.1.1 program [13] (https://sourceforge.net/projects/bowtie-bio/files/bowtie/1.2.0/); the average number of the sRNA mapped and unmapped to host were plotted.”

  1. 3.1. In former Figure 1B, now Figure 2, I suggest using line graphs to display an average of sRNA sizes in libraries to compare sRNA size distribution in the host compared to the virus.

Response: line graph with average of mapped- and unmapped-host reads was included as the Figure 1 in the present version of the manuscript.

  1. 3.1. In former Figure 1, now Figure 2, a short subtitle above each graph would be helpful to determine the results shown in each.

Response: The former Figure 1 is the Figure 3 in the present version of the manuscript. A short subtitle above each graph was included.

  1. 7. 3.2 To better understand the graphical depiction of results in Figure 2, now Figure 3, I suggest dividing Figure 2 into 2 parts, 2A. The figure title should reveal the main finding of the figure. Divide top and bottom panels into A and B and turn each graphic 90º counterclockwise.

Response: The Figure with the heatmap of the viral distribution patterns in domesticated and wild bean samples is the Figure 4 in the present version of the manuscript. The figure was turn 90° counterclockwise, although this was not divided into panels, the number of clusters was included in the dendrogram on the left side. We think that this changes contribute to a better understand and interpretation of the graphic.

Describe the clustering A.

Figure 2 is not described in text. In the text provide a clear description of Figure 2 (Figure 3A and 3B), including methods and results. Include the results and conclusions of the clustering shown in the top panel. What is the relationship between the clustering and the regional distribution shown in the lower section? Please include a description of the conclusion based on the results shown in lower part of the graph. The font used for the library names in the X-axis is too small.

Response: Methods for the heatmap analysis were included in section 2.3: “The distribution patterns in domesticated and wild beans from MCD and Nayarit were analyzed through a heatmap that was built using the function heatmap2 from the package gplots [15] on R 4.0.3 [16]. Code is available in Supplementary Materials.” A clear description and conclusion of Figure 4 was included under the subtitle: “3.3. Viral distribution patterns in domesticated and wild bean” The font size used for the libraries names was increased.

 Discussion

  1. Add subtitles to separate sets of paragraphs in the discussion to identify sections according to the main issues and conclusions addressed.

Response: Subtitles were included to separate the main paragraphs in the discussion

Reviewer 2 Report

In this article, Chiquito-Almanza et al. investigated the virus prevalence in wild and domesticated populations of common bean by using small RNA sequencing data and validated the information by targeted nucleic acid amplification and dideoxy terminator sequencing.   They presented an interesting result of significantly higher viruses prevalence in the domesticated bean relative to that in wild bean with a comprehensive discussion on the result.  Overall,  I think the manuscript is well written and the data provided could be useful for future studies on the emerging viruses in domesticated beans or other crops.   However, I have a few comments/suggestions to the authors, mainly on the method/result sections. 

  • The authors provided information of the 42 samples of wild bean and 30 samples of domesticated bean in the supplementary table (Table S1). I suggest the authors to include more detail labels on this table.  For example, I think the samples with “NA” in column “Cultivar” are samples of wild bean, could the author have a separate column to separate domesticated samples and wild samples?  Also, for the developmental stages (R5-R9), it is better to include more information.
  •  At line 145-147, the authors mentioned that they mixed the RNA from several plants into a bulk sample/library for sequencing. Are those mixed samples closely related or having similar phenotypes?  I wonder how would this affect the prevalence calculation (i.e., the number of samples that contained at least one virus out of the total number of individual samples)?  Here, is the merged sample counted as one individual samples?
  • At line 152, the authors mentioned they used Mann-Whitney U-test to compare the differences of viral prevalence between wild and domesticated beans and showed the data in Figure 1F.  I wonder how the statistics was performed on virus prevalence value (i.e., the number of samples that contained at least one virus out of the total number of individual samples)?  Is the # virus contained used as input data here?  Please provide explicit description here.  
  • In section 2.3, I suggest the authors provided more detailed description about their bioinformatics analyses, including parameters or threshold values used and how they determined if the assembled genome is complete.
  • I didn’t see any place in the main text referring Figure 2. Please include it in the main text.

Author Response

Thank you very much for the comments and suggestions, the answers to your questions are below.

The authors provided information of the 42 samples of wild bean and 30 samples of domesticated bean in the supplementary table (Table S1). I suggest the authors to include more detail labels on this table. For example, I think the samples with “NA” in column “Cultivar” are samples of wild bean, could the author have a separate column to separate domesticated samples and wild samples? Also, for the developmental stages (R5-R9), it is better to include more information.

Response: Table S1 was improved with the recommendations suggested by the reviewer. For each region (Guanajuato, Jalisco and Nayarit) a header with the legend “Samples of domesticated bean” and “Samples of wild bean” was included before the group of samples of domesticated and wild bean, respectively. In the “Type” column the specification “Wild bean” was included for the wild bean samples. A foot note with a brief description of the development stage according to Fernández et al., 1986, and a note related with the stage of development of wild bean were included.

At line 145-147, the authors mentioned that they mixed the RNA from several plants into a bulk sample/library for sequencing. Are those mixed samples closely related or having similar phenotypes?

Response: Each composite library was integrated by three to six plants. The plants of each bulk were collected from the same spot in the population sampled, these had similar phenotypes, so we suppose that are closely related.

I wonder how would this affect the prevalence calculation (i.e., the number of samples that contained at least one virus out of the total number of individual samples)? Here, is the merged sample counted as one individual samples?

Response: Total nucleic acids (DNA and RNA) of each sample used for the construction of the composite libraries was isolate, and later used to confirm by RT-PCR or PCR the viruses identified by sRSA in each single plant that integrated the composite library. Therefore, virus prevalence was calculated considering individual plants and was defined as the proportion of samples taken from individual plants that contained at least one virus.

At line 152, the authors mentioned they used Mann-Whitney U-test to compare the differences of viral prevalence between wild and domesticated beans and showed the data in Figure 1F. I wonder how the statistics was performed on virus prevalence value (i.e., the number of samples that contained at least one virus out of the total number of individual samples)? Is the # virus contained used as input data here? Please provide explicit description here.

Response: Virus prevalence was calculated as the percentage of the number of individual samples with at least one virus divided by the number of total individual samples. To compare the differences of viral prevalence between samples collected from domesticated and wild beans from each region, a Mann-Whitney U-test was perform using as input data the number of samples of individual plants that contained at least one virus from the total of individual samples tested.

In section 2.3, I suggest the authors provided more detailed description about their bioinformatics analyses, including parameters or threshold values used and how they determined if the assembled genome is complete.

Response: For the de novo assembled of reads into contigs and identification of viruses we used the online VirusDetect with default parameters. In section 2.3 more information about the parameters used was included: “The de novo assembled of reads into contigs and identification of viruses were conducted according to Zheng et al. [14] using the online VirusDetect pipeline v1.6 with default parameters (http://virusdetect.feilab.net/cgi-bin/virusdetect/index.cgi); the common bean genome and plant virus reference database in the VirusDetect software were used as a reference to subtract host sRNA and for reference-guided assemblies through aligning sRNA sequences, other parameters applied were coverage = 0.1 and depth = 5.”

We did not perform a completeness calculation for the assembled genomes. The term “completeness” was replace by “coverage %”. A reconstructed genome was considered complete when its sequence coverage 100% of the complete genome used as reference. The follow statement was included in section 2.3: “The percentage of coverage of the reconstructed genome upon the viral genomes used as reference were also determined. The assembled genomes were considered complete if they cover 100% of the complete genome used as reference.”

I didn’t see any place in the main text referring Figure 2. Please include it in the main text.

Response: Figure 2, now Figure 4, was described under the subtitle: “3.3. Viral distribution patterns in domesticated and wild bean” as also solicited by reviewer 1.

Reviewer 3 Report

It was nice research about the diversity and distribution of viruses infecting in Phaseolus spp.  And great collected information different viruses of beans.

I recommend to authors prepare table/tables to summarize result which needs to include;

Virus-Abbreviation and taxonomy-Genome type-Total number of sequence-Complete sequence-GenBank accession #- Symptoms of virus-Transmission of virus-   

Author Response

Thank you very much for the comments and suggestions, the answers to your questions are below.

It was nice research about the diversity and distribution of viruses infecting in Phaseolus spp. And great collected information different viruses of beans.

I recommend to authors prepare table/tables to summarize result which needs to include;

Virus-Abbreviation and taxonomy-Genome type-Total number of sequence-Complete sequence-GenBank accession #- Symptoms of virus-Transmission of virus-

Response: Table 1 with virus names, acronym, genus, family, genome type, total number of complete or near complete sequences of assembled genomes, and transmission of identified viruses was included. The GenBank accession numbers, and symptoms of each sampled plant are included in Supplementary Table 1.

Round 2

Reviewer 2 Report

The authors have kindly well addressed all my questions and comments.  I agree to have this manuscript to be finally published on the Journal.